# Microclimatic Evaluation of Five Types of Colombian Greenhouses Using Geostatistical Techniques

**DOI:** 10.3390/s22103925

**Published:** 2022-05-22

**Authors:** Edwin Villagrán, Jorge Flores-Velazquez, Mohammad Akrami, Carlos Bojacá

**Affiliations:** 1Department of Biological and Environmental Sciences, Faculty of Natural Sciences and Engineering, Jorge Tadeo Lozano University, Bogotá 111321, Colombia; carlos.bojaca@utadeo.edu.co; 2Corporación Colombiana de Investigación Agropecuaria—Agrosavia, Centro de Investigación Tibaitata, Km 14, vía Mosquera-Bogotá, Mosquera 250040, Colombia; 3Coordination of Hydrosciences, Postgraduate Collage, Carr Mex Tex km 36.5, Montecillo Edo de Mexico 62550, Mexico; jorgelv@colpos.mx; 4Department of Engineering, University of Exeter, Exeter EX4 4QF, UK

**Keywords:** passive greenhouses, spatial variability, microclimatic heterogeneity, microclimate optimization, semivariogram

## Abstract

In Colombia, the second-largest exporter of cut flowers worldwide and one of the South American countries with the largest area of crops under cover, passive or naturally ventilated greenhouses predominate. Locally, there are several types of greenhouses that differ in architecture, size, height, shape of roof and ventilation surfaces, of which many characteristics of the microclimate generated in their interior environment are unknown. This generates productive limitations that in some way may be limiting the yield, quality and health of the final products harvested; in addition, Colombian producers do not have the ability to monitor the microclimate of their farms, much less to correlate microclimate data with data on crop production and yield. Therefore, there is a need for the Colombian grower to know the most relevant microclimate characteristics generated in the main greenhouses used locally. The objective of this work was to carry out a microclimatic characterization of the five most used types of greenhouses in Colombia. The main results allowed determining that in these structures, there are conditions of high humidity and low vapor pressure for several hours of the day, which affects the physiological processes of growth and development of the plants. It was also identified that for each type of greenhouse, depending on the level of radiation, there is a significant microclimatic heterogeneity that may be the cause of the heterogeneity in plant growth, which is a common characteristic observed by the technical cultivation personnel. Therefore, it can be concluded that it is urgent to propose microclimatic optimization strategies to help ensure the sustainability of the most important production systems in the country.

## 1. Introduction

In crops grown under protective environment structures, microclimate management is undoubtedly one of the most relevant factors affecting crop production in terms of both quality and quantity [1,2]. Worldwide, there are more than 5.2 million hectares covered with different types of protective agriculture structures [3,4]. These structures present different technological levels, from a low technological level, such as that found in the small tunnels or structures typically used in tropical countries of Latin America and Asia [5], to an extremely high level, such as that found in the high-tech greenhouses found mainly in northern European countries, where it is possible to fully manage the variables that affect crop growth and development [6].

In Colombia, there is an important area dedicated to cultivation under cover, mainly for the production of cut flowers for the international market; locally, different types of cover structures are built, mostly with plastic covers and made of low-cost materials, such as wood and non-structural steel [7]. These types of structures, known locally as passive greenhouses, are also used for the production of vegetables, aromatic species and medicinal herbs [8]; additionally, they are established in different climatic regions of the country without considering that their design is mainly for regions where cold climates predominate [9].

Colombian structures present diverse shapes that differ according to the criteria of the greenhouse producer or builder. Therefore, aspects such as the height, geometric shape, width and length of the structure are quite diverse, as well as the areas and arrangement of the ventilation surfaces [7]. All the above-mentioned characteristics directly influence the natural ventilation phenomenon, which is the main method of climate control in passive-type greenhouses [10]. Therefore, the microclimate generated on a spatial and temporal scale inside these structures is directly related to the ventilation rate and to the distribution of airflow patterns generated by wind or thermal effects [11].

At the local level, there is no culture on the part of the producer for the recording and correlation of microclimatic data with production data. Likewise, the agronomic management of the plants and the management of natural ventilation in the structure are done through raw data recorded through a single sensor located in the central area of the greenhouse. This is undoubtedly a negative aspect, even more so when it is known that for the same temporal moment, the microclimate conditions usually present different values in magnitude in different positions inside the greenhouse [12]. This culture of the Colombian producer can undoubtedly be changed and improved, which would allow optimizing greenhouse production at the local level. Since the monitoring of the greenhouse microclimate in an agile and precise way will facilitate the analysis of the information in a multidimensional way and its application to the practices and cultivation work, this will improve the efficiency in the use of physical resources [13].

The characterization of the microclimate in greenhouses is one of the areas of greatest interest for researchers dedicated to the study of crops under cover. Among the analysis methodologies is the implementation of different physical and mathematical models and the use of various computer simulation tools that allow the spatial distribution of the microclimate inside greenhouses or other protected agriculture structures to be determined [9,14,15]. Other less-common methodologies are based on the installation of sensor grids that allow monitoring and recording spatial differences in temperature and relative humidity for the same time scale [16]. This climatic heterogeneity in greenhouses is a negative aspect and affects the uniformity of crop production since it directly affects the transpiration and nutrient absorption rates of the cultivated species [17,18,19].

In Colombia, it is possible to improve the micro-climatic behavior of the main greenhouse structures by implementing passive climate control strategies for both daytime and nighttime conditions since the use of high-tech greenhouses does not represent a viable way forward due to the high initial investment cost and the high energy consumption characteristic of this type of greenhouse [3]. Likewise, the increase in productive yields that can be expressed by the main ornamental crops in high-tech structures in tropical climates is still unknown. In order to answer this last question, it is possible to seek answers through the use of simulation models, such as the one proposed by Vanthoor et al. [20], although this is not the main objective of this work. For daytime hours, it is necessary to improve ventilation rates and airflow patterns, seeking to increase the renewal rates so that the management of ventilation areas can regulate the excess temperature and humidity and also maintain CO_2_ levels at values similar to those of the indoor environment, it is also possible to use shading screens for times of high radiation [9,21]. While for the night hours, strategies to reduce the loss of infrared thermal radiation can be considered, it is also necessary to increase the level of thermal tightness of the greenhouse through the use of energy-saving thermal screens or specialized roofing plastics, and finally, energy storage systems and ground-air heat exchangers could be designed and implemented [22,23].

A microclimatic optimization of Colombian greenhouse structures aiming at increasing the yields of the main crops under cover must be preceded by a characterization of the microclimate behavior of the most-used greenhouse models at the local level. This information will allow us to establish when and how much we should manage and control variables such as temperature and air humidity. Therefore, the objective of this work was to characterize the microclimate behavior of the five most-used greenhouses in Colombia. The methodology consisted of measuring variables such as temperature, humidity, vapor pressure deficit (VPD), internal solar radiation level and CO_2_ concentration inside the greenhouse on an hourly scale. Finally, using a grid of 40 temperature and humidity sensors and implementing geostatistical techniques, a spatial variability analysis was carried out to determine the climatic heterogeneity present in this type of greenhouse.

## 2. Materials and Methods

### 2.1. Description and Location of the Greenhouses

For this study, 5 types of greenhouses built in the savanna of Bogota were selected, among which there were already well-developed ornamental crops of Rosa and Carnation. Each of the analyzed greenhouse models presents variations in their shape, size, surface and arrangement of the ventilation areas. The main characteristics of each greenhouse are described below. 

#### 2.1.1. Traditional Greenhouse (TG)

The traditional Colombian greenhouse (TG) is one of the most widely used models in the country for flower production, as well as for vegetable and aromatic production [8]. The greenhouse evaluated was located in the municipality of Guasca, department of Cundinamarca (4°50′57.60′′ N, 73°52′58.19′′ N and altitude: 2664 m.s.a.l). 

The structure of the TG greenhouse is made of wood, and its cover is made of commercial polyethylene of 0.2 mm thickness, the management of the microclimate is performed through the management of the ventilation areas arranged on the 4 sides of the greenhouse and in the region of the roof, the technical specifications and some architectural details of TG can be consulted in Table 1.

#### 2.1.2. Multi-Tunnel Greenhouse (MG)

Another commonly used greenhouse is the multi-tunnel (MG) type; the model evaluated was located in the municipality of El Rosal, department of Cundinamarca (4°50′56.27′′ N, 74°14′16.72′′ N and altitude: 2592 m.s.a.l). This type of greenhouse is manufactured with a metallic structure, and the cover is made of commercial plastic of 0.2 mm thickness, the management of the microclimate is performed through the management of the ventilation areas arranged on the sides and a region of the roof, the main characteristics of MG can be reviewed in Table 2.

#### 2.1.3. Hanging Type Greenhouse (HG)

To a lesser proportion, we find greenhouses of the so-called hanging type (HG), this type of structure is made of wood and with a commercial plastic cover of 0.2 mm thickness. The structural form of HG presents a structural variation with respect to the traditional TG greenhouse in order to obtain higher thermal inertia and a lower level of shading, based on a larger volume of air enclosed in the roof area and a smaller number of structural elements with respect to TG. In this study, two models of hanging greenhouses were evaluated, HGT1, which had a larger covered area than HGT2; both HGT1 and HGT2 had ventilation surfaces with a similar opening on all four sides of the greenhouse and in the roof area. On the other hand, another marked difference between HGT1 and HGT2 is related to the arrangement and orientation of the pivot roof windows, which, in the case of HGT1, are all oriented in the same direction and arranged in the same region of each span, while in HGT2, the pivot windows have alternate orientations and different arrangements in each span. The greenhouses evaluated were located in the municipality of Nemocón, department of Cundinamarca. (5°03′26.44′′ N, 73°55′47.34′′ N and altitude: 2570 m.s.a.l) and some general characteristics are summarized in Table 3 and Table 4.

#### 2.1.4. Spatial Greenhouse (SG)

Finally, a spatial greenhouse (SG) was evaluated. This type of structure is built with concrete pillars, elements that function as columns, while the roof region is made of intertwined steel cables on which the plastic covers are fastened, and a ventilation area is generated. This type of greenhouse is equipped with ventilation surfaces on the 4 sides and in a region of the roof of each span. The model evaluated was located in the municipality of Soacha, department of Cundinamarca. (4°35′35.53′′ N, 74°12′32.70′′ N and altitude: 2546 m.s.a.l) and its main characteristics are shown in Table 5.

### 2.2. Monitoring and Recording of Microclimatic Data

Table 6 lists all of the meteorological measuring equipment used both in the outdoor environment and inside each of the greenhouses. Inside each of the greenhouses, it was decided to monitor the values of temperature, relative humidity, vapor pressure deficit (VPD), radiation level and CO_2_ concentration level since these are the main microclimate variables that affect the growth and development of crops inside protected agriculture structures [24,25]. Generally, the presence of the translucent cover and the geometry of the greenhouse cause the temperature and humidity inside the greenhouse to increase, the radiation level to decrease, and the CO_2_ level to fluctuate more in the time scale, so it is expected that these variables will affect the quality and quantity of the harvested product [26]. Finally, it is also important to mention that, in terms of microclimate and crop growth simulation, the variables studied in this research are the most predicted from any simulation methodology since they are directly related to agricultural production and energy and resource efficiency inside a greenhouse [27].

The pyranometer and the CO_2_ sensor inside the greenhouse were placed in the central zone of each structure, while the 40 thermocouples were evenly distributed according to the covered area of each greenhouse; the 40 sensors were included inside a ventilated capsule that was placed 1.6 m above the ground level just above the zone where the plant canopy grows. Each of the greenhouses was monitored for 40 calendar days (Table 7), and the frequency of data recording was established every ten minutes; these data recording periods are comparable since climatic conditions in Colombia do not vary greatly throughout the months of the year. The vapor pressure deficit variable was determined by means of psychrometric Equations (1)–(3) that relate air temperature and relative humidity, as described in the work developed by Huang et al. [28].
(1)$$VPD=e_{s}-e_{a}$$
(2)$$e_{a}=\frac{HR\times e_{s}}{100}$$
(3)$$e_{s}=0.61078\times \frac{e^{17.2694\times T_{m}}}{T_{m}}$$
where ($e_{s}$) is the saturation vapor pressure and ($e_{a}$) is the actual vapor pressure.

### 2.3. Geostatistical Analysis

The analysis of the collected data using geostatistical techniques was carried out in three phases, the first of which is known as exploratory analysis, the second phase as structural analysis and the third and last phase is the one that allows us to generate the spatial prediction curves of the variables of interest and is known as the prediction phase [29]. To perform the geostatistical analysis, the datasets were organized, grouped and averaged according to 5 intervals categorized according to the outdoor solar radiation level (ESRL). These radiation levels were determined from exploratory analyses of the data, and an attempt was made to group the data into radiation intervals representative of the study region, as performed by Bojacá et al. [12] (Table 8). In these types of structures, the spatial variability of variables such as temperature, relative humidity and VPD is dependent on the level of radiation; therefore, more specific analyses should focus on what happens in the daytime period, the period between 6 and 18 h of the day (ERL 2–ERL 5), while at night, the spatial variability of the variables is less evident due to the lack of solar radiation in the nighttime period between 18 and 6 hours of the following day (ERL 1).

#### 2.3.1. Exploratory Analysis Phase

This is performed with the objective of verifying that the datasets comply with some geostatistical assumptions, such as stationarity and seasonality. Usually, the graphs are generated to inspect the stationarity of the study variables [30]. Additionally, it is verified if the datasets are seasonal; that is, if they present a periodic behavior, for it is assumed that the degree of association of the data depends on the spatial separation and not on the location of the data in space [31]. Finally, basic statistics, such as the Shapiro–Wilk test, were calculated to determine the normal distribution of the data, which, together with the initial graphical analysis, allow us to rule out the existence of any trend in space [12]. In this work, these calculations were performed on an hourly scale for temperature, relative humidity and vapor pressure deficit. On the other hand, radiation and CO_2_ concentration data were also averaged both indoors and outdoors, although for these two variables, the geostatistical analysis is not complete since there is only one data recording point for each variable inside the greenhouses.

#### 2.3.2. Structural Analysis Phase

In this phase, the empirical semivariograms associated with each dataset of each variable collected must be constructed, which is nothing more than the graphical relationship that exists between the spatial separation distances and the semivariance of the 3 variables studied [32]; this semivariogram was constructed by means of the equation
(4)$$\gamma \left(d\right)=\frac{1}{2Mp\left(d\right)}\sum_{i=1}^{Mp\left(d\right)}{\left[W\left(xi\right)-W\left(xi+d\right)\right]}^{2}$$
where $Mp_{d}$ is the number of pairs at distance d, d is the increase, $W\left(xi\right)$ is the experimental values and xi is the locations where the values are measured W(xi). Empirical semivariograms should be fitted to theoretical semivariogram models; in this case, we used and evaluated the fit for 5 theoretical models (Gaussian, circular, spherical, Cauchy and Exponential). These models have the following parameters: range (*a*), sill (*c*_1_) and nugget (*c*_0_)*_._* The nugget represents both sampling error and spatial variability for the minimum distance interval, the sill represents the upper asymptote of the semivariogram and corresponds to the maximum variability explained by the semivariogram, and the range represents a measure of the maximum distance of influence [12]. The theoretical semivariogram model to be used in the prediction phase was selected according to the Bayesian information criterion (BIC) and the Akaike information criterion (AIC). The model with the lowest values for these criteria will be the model that directly presents the best fit to the experimental data.

#### 2.3.3. Prediction Phase

The information from the two previous phases is used to predict the behavior of the variables of interest at unsampled points based on experimentally sampled values and the calculation of their spatial continuity structure [33]. Likewise, the predicted values are used to create the spatial distribution curves in a two-dimensional plane at a height of 1.6 m above ground level. The selected prediction method is ordinary kriging, which is considered to be an optimal linear predictor, unbiased and with minimum variance. The prediction is calculated with the following equation
(5)$$Z\left(x_{0}\right)=\overset{m}{\underset{i=1}{\sum }}\lambda_{i}W\left(x_{i}\right)\;\;\;\;\;\;\;\;with\;\;\;\;\;\overset{m}{\underset{i=1}{\sum }}\lambda_{i}=1$$
where $\lambda_{i}$ is the weight assigned to each value of the variables in the observed positions *W*(*x*_0_) based on the parameters of the previously adjusted theoretical semivariogram.

## 3. Results and Discussions

### 3.1. Exploratory Analysis

For the analysis of each of the variables, the data were averaged on an hourly scale for the total number of sensors used both indoors and outdoors of the greenhouse and for the total number of days of the measurement.

#### 3.1.1. Temperature

Figure 1 shows the average temperature of all sensors on an hourly scale inside the greenhouse and the hourly average value of the outside sensor. In general terms, the typical behavior of this type of structure in tropical climates is observed for all the greenhouses evaluated, with the lowest temperature conditions occurring at night and the highest between noon and 15:00 hours, just after the moments of maximum external solar radiation [23]. The mean temperature (T_m_) values for the 24 h inside the greenhouses were 15.1 ± 3.6, 14.0 ± 3.5, 15.9 ± 5.1, 15.7 ± 3.5 and 15.9 ± 3.1 °C for TG, MG, SG, HGT1 and HGT2, respectively, which are 2.0, 0.9, 1.6, 1.4 and 1.8 °C higher than the outside ambient conditions for each of these greenhouses.

On the other hand, the maximum mean values (T_mh_) occur just between 11 and 14 h for TG, MG, SG, HGT1 and HGT2, where values of 21.4, 19.4, 24.3, 21.5 and 21.1 °C, respectively, were obtained. These T_mh_ values, with the exception of SG, are not within the ranges established as optimal daily temperature values, which should range between 24 and 30 °C for the optimal growth and development of a great variety of ornamental and vegetable species cultivated in greenhouses [4]. For the case of the minimum mean values (Tml), these values were 11.3, 9.8, 10.1, 11.8 and 12.7 °C, representing thermal differentials between the indoor and outdoor environment of 1.1, 0.03, −0.7, 0.07 and 1.7 °C, for TG, MG, SG, HGT1 and HGT2, respectively. It is important to highlight that the values in all the greenhouses are below the minimum value of optimum temperature for species such as Rosa and Carnation, which, according to the literature, should range between 14 and 16 °C [34]. 

Finally, it is also important to mention that for MG and SG, we can observe, in Figure 1 and in the thermal differentials, values that show that in the interior of these greenhouses, in the hours of the night and the dawn of 20 and 5 hours, the phenomenon of thermal inversion occurs, a phenomenon where the greenhouse temperature is lower than the outside environment [35]. This phenomenon undoubtedly affects crop growth rates and is caused by the ineffectiveness of these greenhouses in retaining the loss of thermal radiation that occurs from the greenhouse floor to the outside environment, as well as the poor hermeticity of these local structures [36]. Therefore, for these conditions, it is imperative to establish climate control strategies to optimize the climate under nocturnal conditions in these types of greenhouses used in Colombia, as mentioned in previous studies [36,37].

#### 3.1.2. Relative Humidity

In the case of relative humidity (RH), the hourly behavior can be observed for each greenhouse in Figure 2. This variable is undoubtedly the most difficult to control within the greenhouse microclimate management since its value will depend on the psychrometric condition of the air and the mass contributions that occur via transpiration and water evaporation phenomena inside the greenhouse [4,38]. It can be observed that the humidity values in the greenhouses always show values higher than 67% for the average hourly value; the maximum humidity conditions inside the greenhouses occur during the night hours and generally exceed 90%, while the minimum values occur between 11 and 14 h and range between 67% and 75%.

The RH value for the 24 h of the day was 93 ± 6%, 87 ± 7%, 84 ± 11%, 88 ± 11% and 87 ± 12% inside TG, MG, SG, HGT1 and HGT2, respectively, values differing by 6%, 2%, 6%, –1.4% and –0.5% with respect to the outside environment for each greenhouse. Therefore, it is evident that for TG, MG and SG, the RH conditions are higher than the outdoor environment, and the RH values inside the greenhouses are above the RH values recommended for crops under cover, which range between 60 and 80% [39]. These conditions of high humidity inside the greenhouses generate a favorable environment for ornamental or horticultural species to be infested and affected by pathogens that generate fungal diseases, which have a high capacity to cause damage to plants. Therefore, productive losses and environmental damage associated with the application of chemical control substances for diseases, such as gray mold and downy mildew, caused by *Botrytis cinérea* and *Chromista Peronospora sparsa*, occur [40,41]. 

#### 3.1.3. Vapor Pressure Deficit (VPD)

One of the variables of interest for decision-making in climate and irrigation management inside the greenhouse is the VPD, a variable that relates the difference between the water vapor pressure at saturation and the actual water vapor pressure at a pressure from the greenhouse temperature [39]. VPD is a variable dependent on the temperature and RH of the greenhouse environment; its importance for climate management lies in the fact that this variable is the driving force of plant transpiration [4]. Its behavior in each of the greenhouses can be observed in Figure 3; it is observed that the maximum values of VPD occur during daytime hours, precisely in the periods where the temperature increases and the RH decreases, while the lowest values of VPD occur during the nighttime period, where the cold and saturated environments predominate in each of the greenhouses.

The mean values of VPD for the 24 h of the day were 0.12 ± 0.12, 0.24 ± 0.17, 0.36 ± 0.34, 0.23 ± 0.24 and 0.24 ± 0.26 kPa, in TG, MG, SG, HGT1 and HGT2, respectively, values that are lower by 0.08, 0.02, and 0.03 kPa for TG, MG and SG with respect to the outside environment, while HGT1 and HGT2 values are higher with respect to the outside environment value by 0.04 and 0.02 kPa, behaviors that are directly related to the thermal and humidity conditions present in each greenhouse. The maximum mean VPD values in the greenhouses were 0.34, 0.5, 0.98, 0.70 and 0.71 kPa for TG, MG, SG, HGT1 and HGT2, respectively. These VPD values, except for TG, were during the 11 and 15 hours within the recommended ranges for plant growth, where VPD should range between 0.45 and 1.25 kPa, although the ideal should be at 0.9 kPa [42]. These values present in greenhouses undoubtedly limit the productive potential of crops, as VPD affects photosynthetic activity and plant nutrition, causing deficiencies due to limited sap flow movement [43,44].

In the case of the minimum mean VPD values, these values were 0.001, 0.06, 0.05, 0.01 and 0.008 kPa for TG, MG, SG, HGT1 and HGT2, respectively. These values, so close to 0, generally occur during the entire night period and are quite inadequate for plant growth, generating a favorable environment for the appearance and survival of fungal diseases [45,46]. Therefore, it is necessary to analyze these situations and look for alternatives to improve the microclimate conditions related to VPD present in the greenhouses evaluated. 

#### 3.1.4. Solar Radiation

Solar radiation is the energy engine of the greenhouse and is the variable responsible for generating the greenhouse effect inside the structures and its respective increase in temperature; also, part of this energy is captured by the soil to be liberated at night [47]. The hourly behavior of solar radiation can be seen in Figure 4. In general, it is observed that the radiation value increases from 6 to 13 hours and then decreases again between 13 and 18 hours, this being a typical behavior of the value during the daytime period in tropical regions.

The mean radiation values obtained during daylight hours were 180 ± 142, 235 ± 159, 224 ± 177, 145 ± 115 and 144 ± 108 Wm^−2^ for TG, MG, SG, HGT1 and HGT2, respectively. These values represent a transmission percentage of 75%, 87%, 76%, 74% and 71% for each of the greenhouses mentioned above. It should be noted that the roof that generates the highest radiation transmission is the tunnel type of the MG greenhouse; unlike the flat roofs of the other greenhouses, this percentage of transmission depends on the roof material, which was similar for all greenhouses in material and age of use, while the geometric shape of the roofs was different and this generates a different percentage of transmission [48].

#### 3.1.5. Carbon Dioxide (CO_2_) Concentration

Finally, the concentration of carbon dioxide (CO_2_) is another microclimatic variable that directly affects the photosynthetic assimilation of plants and, therefore, is a variable that must be monitored inside greenhouses [49]. In the case of this study, it should be mentioned that the greenhouses studied do not have carbon enrichment systems, and there is no local information on the behavior of this variable in these types of greenhouses. Figure 5 shows the hourly trend of the CO_2_ concentration in each of the greenhouses, the mean values of CO_2_ concentration in the greenhouses were 381.4 ± 66.4, 396.2 ± 57.5, 391.7 ± 42.6, 397.7 ± 66.1 and 409.4 ± 48.9 ppm for TG, MG, SG, HGT1 and HGT2, respectively. 

The maximum values of CO_2_ concentration occur during the night hours, mainly due to the contribution of CO_2_ to the greenhouse environment by the plants through the process of nocturnal respiration. The mean CO_2_ concentration values were 463, 479, 460, 502 and 481 ppm in TG, MG, SG, TGH1 and TGH2, respectively. Subsequently, with the appearance of the sun and the beginning of the photosynthetic process of the plants, the CO_2_ level in the greenhouses decreases to minimum levels around 15 and 16 CO with values of 290, 321, 336, 340 and 346 ppm in TG, MG, SG, TGH1 and TGH2; values below the CO_2_ concentration in the outside environment.

This is due to the fact that plants are consuming CO_2_ in their photosynthesis process at a faster rate than the CO_2_ renewal rate of the air via natural ventilation; therefore, this factor may be indicating that the ventilation rates in greenhouses are not adequate [50]. The relationship between the level of CO_2_ concentration in the greenhouse air and its ventilation rates has been demonstrated [51], and in this study, it can be verified since it was found that the greenhouses with the largest ventilation surfaces (HGT1 and HGT2) are the ones with CO_2_ curves in the diurnal period that are very close for both outside and inside environments.

#### 3.1.6. Testing of Geostatistical Assumptions

In order to rule out any kind of trend in the datasets collected and, additionally, to verify that these same datasets fit a normal distribution, the graphs shown in Figure 6 are constructed. In this case, trends can be observed for data collected in MG and ESRL 5 for temperature (Figure 6A), in ESRL 3 for relative humidity (Figure 6B) and in ESRL 4 for VPD (Figure 6C). These curves show that for both the width and length of the greenhouse, most of the values of the three variables analyzed are randomly distributed without any trend; the temperature ranges between 17.8 and 22.0 °C, the relative humidity between 70% and 96% and the VPD between 0.01 and 0.86 kPa.

It is also possible to infer, in principle, that from the bell-shaped histograms of the three variables, there is an adjustment to the normality of the dataset. This was confirmed by calculating the Shapiro–Wilk test, where it is established that to guarantee the normality of the data, the *p*-value must be greater than the defined alpha (0.05), and for these datasets, *p*-values of 0.48, 0.39 and 0.33 were obtained for temperature, relative humidity and VPD, respectively. This normality of the datasets is always desirable in this type of work since it guarantees adequate predictions of the study variables by means of the kriging method [52]. This same procedure was performed for all 75 datasets collected in this study, obtaining the same satisfactory results as for the 3 datasets discussed above.

### 3.2. Structural Analysis 

Table 9 shows the values obtained for 75 datasets corresponding to the analysis of the fit of the experimental data to theoretical semivariogram models. In general terms, it can be observed that as ESRL 5 radiation levels increase, there is a greater variation in temperature, humidity and VPD inside the greenhouses; on the other hand, when there is a lower level of ESRL 1 radiation, these microclimatic variations inside the greenhouses are less significant. For the case of temperature, it was found that for 19 datasets, the best fitting theoretical semivariogram model (lowest AIC and BIC value) was the circular model, followed by the spherical and Gaussian models each with 3 datasets, respectively. In the case of relative humidity, in 20 datasets, the best theoretical semivariogram model was the circular model, and in 5 datasets, the Gaussian model; finally, for VPD, 24 datasets showed a better fit for the circular semivariogram model and only 1 dataset under the Gaussian model. 

The values of the nugget or nugget effect (*c*_0_) for temperature ranged from a maximum value of 0.37 in MG and ESRL 5 to a minimum value of 0.0 obtained in 11 of the 25 datasets, in the case of relative humidity; these values of *c*_0_ ranged from a maximum value of 6.9, obtained in SG and ESRL 5, to a minimum value of 0.0 obtained in 15 of the 25 datasets. Finally, in VPD, the value of the nugget effect ranged between a maximum of 1.88, obtained in HGT2 and ESRL 2, and a minimum of 0.0, obtained in 21 of the 25 datasets for this variable, all these values of *c*_0_ can be considered low, which allows us to deduce that the spatial dependence due to the error of the experimental measurement is not relevant [12]. For the sill parameter (*c*_1_), the values for temperature ranged from a minimum value of 0.09, obtained in TG and ESRL 1, to a maximum of 3.1, in SG and ESRL 5. In the case of relative humidity, the values of *c*_1_ ranged from a minimum value of 0.0, obtained in HGT1 and ESRL 1, to a maximum value of 114.1 in SG and ESRL 5. For VPD, the minimum value was 0.0, obtained in several datasets, and the maximum value was 0.22, obtained in SG and ESRL 5. In general terms, it was found for the three variables that as the level of radiation increases (ESRL 1 to ESRL 5), the values of *c*_1_ increase, which allows us to affirm that as the level of radiation increases, a greater spatial variation of temperature, relative humidity and VPD is found inside the greenhouses [41]. 

For the ratio (*c*_0_/*c*_1_), it was found that the values for temperature ranged from minimum values of 0.0% obtained in several datasets to a maximum of 18.6% obtained in MG and ESRL 4; for humidity the minimum value was 0.0%, obtained in several datasets, and a maximum value of 22.1% obtained in TG and ESRL 5, while for VPD, the minimum value was 0.0% in most of the datasets and a maximum of 4.5% in SG and ESRL 4. These values, being less than 25%, can be considered adequate and are indicative of a strong spatial dependence between the data [53]. These values also allow us to conclude that the number of sensors used in each of the greenhouses is sufficient for the prediction of temperature, relative humidity and VPD values in the unsampled points of each greenhouse [12].

### 3.3. Prediction of the Spatial Behavior of the Microclimate

The prediction of the variables analyzed for non-sampled points inside the greenhouse was carried out using the ordinary kriging method. These estimations are a function of the results of the adjustment of the theoretical semivariograms, so their validity is given by the quality of the adjustment of the theoretical semivariograms.

For the analysis of spatial behavior, we selected the contrasting ESRLs (ESRL 1 and ESRL 5), which will allow us to analyze the scenarios of greater and lesser microclimatic homogeneity. The other scenarios (ESRL 2, ESRL 3 and ESRL 4) are included graphically in Appendix A of this document.

#### 3.3.1. Spatial variability of microclimate in MG

For the case of the aggregated data for the night hours (ESRL 1), it was found that for the MG greenhouse, the value of Tm was 11.1 ± 1.1 °C (Figure 7). It was observed that inside the greenhouse, the temperature in the largest amount of covered area ranges between 10.4 and 11.3 °C, while the maximum magnitude of the thermal differential inside the greenhouse (Δ_TG_) was 4.1 °C between small areas located in the central region and some points of the front sides. In the case of RH (Figure 7), the mean value was 93 ± 1.6 °C, with a very important area between values of 94% and 97%, while the maximum humidity differential (Δ_RHG_) was 11% between some perimeter points of MG. For the VPD variable, the mean value was 0.09 ± 0.03 kPa, while the VPD differential (Δ_VPDG_) was 0.16 kPa between areas located in the center and the perimeter of MG.

The spatial behavior of temperature in ESRL 5 shows a greater climatic heterogeneity in MG (Figure 8). The Tm value was 19.2 ± 1.8 °C, while the Δ_TG_ value was 6.2 °C. On the other hand, the mean RH value was 76 ± 11% with a Δ_RHG_ value of 42%, finally, the mean VPD value was 0.52 ± 0.29 kPa and the Δ_VPDG_ value was 1.05 kPa. 

#### 3.3.2. Spatial Variability of Microclimate in TG

For TG of the night hours (ESRL 1), it was found that Tm presents a value of 11.5 ± 0.9 °C, while the value of Δ_TG_ was 0.8 °C, which can be considered a highly homogeneous value. In the case of RH and VPD, the mean values were 99 ± 0.2% and 0.006 ± 0.04 kPa, respectively, while the values of Δ_RHG_ and Δ_VPDG_ were 0.8% and 0.01 kPa (Figure 9). For ESRL 5, the values of T_m_, RH and VPD were found to be 23.1 ± 2.3 °C, 89 ± 7% and 0.41 ± 0.31 kPa, while the values of Δ_TG_, Δ_RHG_ and Δ_VPDG_ were found to be 6.5 °C, 10% and 0.54 kPa (Figure 10). In the case of temperature, it was observed that there were three quite differentiated areas in magnitude of the energy value in the lateral, central and frontal zones of the TG, while for humidity and VPD, areas with different values were observed in different regions of the TG.

#### 3.3.3. Spatial Variability of Microclimate in SG

For SG, in regards to the spatial distribution of the microclimate in ESRL 1, it was found that some patches appear with a differentiated condition for temperature, relative humidity and VPD; these patches extend in the longitudinal direction of the greenhouse and appear in the same positions distributed along the width of the structure (Figure 11). For this condition, the T_m_, RH and VPD values were found to be 11.6 ± 0.4 °C, 96 ± 3% and 0.07 ± 0.02 kPa, respectively, while the Δ_TG_, Δ_RHG_ and Δ_VPDG_ values were 1.4 °C, 15.1% and 0.13 kPa, although these extreme differential values only appear in small areas of SG.

For the case of ESRL 5, it was mainly observed that the areas of differing magnitude in the microclimate variables expand within SG, increasing the climatic heterogeneity of the greenhouse (Figure 12). The T_m_, RH and VPD values were found to be 24.1 ± 3.9 °C, 72 ± 11% and 0.73 ± 0.35 kPa, respectively, while the Δ_TG_, Δ_RHG_ and Δ_VPDG_ values were 7.2 °C, 39% and 1.6 kPa, with extreme values appearing just above the central zone in SG.

#### 3.3.4. Spatial Variability of Microclimate in HGT1

The spatial behavior of the microclimate for the nighttime hours can be seen in Figure 13. Regions with a differential value in temperature, humidity and VPD were identified in various areas of the interior of HGT1, although it should be noted that the magnitude of the differentials appears to be low. The values of T_m_, RH and VPD were 12.5 ± 0.12 °C, 99 ± 0.2% and 0.0018 ± 0.0011 kPa, respectively, while the values of Δ_TG_, Δ_RHG_ and Δ_VPDG_ were 0.25 °C, 0.3% and 0.0021 kPa. According to these values, it can be considered that the microclimatic condition in this greenhouse is highly homogeneous [54].

For the case of ESRL 5, it is again observed that under this level of radiation inside the greenhouse, there is greater microclimatic heterogeneity (Figure 14). Under this condition, the values of T_m_, RH and VPD were 22.4 ± 1.6 °C, 89 ± 5% and 0.28 ± 0.06 kPa, respectively, while the values of Δ_TG_, Δ_RHG_ and Δ_VPDG_ were 2.7 °C, 3.8% and 0.15 kPa.

#### 3.3.5. Spatial Variability of Microclimate in HGT2

In this greenhouse for the night hours, it was found that the values of Tm, RH and VPD were 13.2 ± 0.17 °C, 99 ± 1% and 0.09 ± 0.05 kPa, respectively, while the values of ΔTG, ΔRHG and ΔVPDG were 0.3 °C, 1% and 0.03 kPa. Further, this greenhouse, under this condition can be considered to present a homogeneous microclimate (Figure 15). In the case of ESRL 5 for HGT2, some differentiated areas of microclimate were found, although to a lower percentage than in the greenhouses analyzed above (Figure 16). For this case, the values of T_m_, RH and VPD were 22.1 ± 0.7 °C, 86 ± 5% and 0.34 ± 0.12 kPa, respectively, while the values of Δ_TG_, Δ_RHG_ and Δ_VPDG_ were 2.8 °C, 12% and 0.34 kPa. 

Finally, these results obtained from the spatial variability analysis allowed a close relationship between the level of solar radiation and thermal heterogeneity inside the greenhouses to be identified; the higher the level of radiation, the greater the magnitude of thermal gradients inside the greenhouses and the greater the area of the internal surface of the structures, results that had been reported by Bojacá et al. [12]. Even this thermal heterogeneity in such naturally ventilated greenhouses has been reported by computational fluid simulation analysis [55]. On the other hand, it was also a common characteristic to find environments with quite high humidity contents and with VPD values very close to 0, both during the day and at night. This is undoubtedly a factor that may be limiting the yield of the crops grown in these greenhouses due to the negative implications that high-humidity environments have on plant growth.

In the same way, it is important to remember that this type of behavior of non-homogeneous microclimates generates differentiated yields in the crops established in these greenhouse structures [56,57]. Even this climatic heterogeneity affects plant breeding research experiments conducted in small greenhouses, causing noise and uncertainties in growth data collected at sampling points inside the greenhouse [58]. Therefore, at the local level, the technical managers of the farms observe accelerated production cycles in some areas within the same greenhouse, as well as low yields and poor-quality products in other regions of the structure [3]. This is even more critical because the management of cultural, agronomic, irrigation and fertilization tasks are established for a greenhouse, assuming that the microclimate within the whole area is homogeneous [59]. Finally, in accordance with the work developed by Fatnassi et al. [60], this type of micro-climatic behavior allows the establishment of pests and diseases that generate large economic losses and can generate a high rate of reports of pest interceptions at the control points located at the ports of export, which are monitored by the sanitary organizations of each country, affecting the phytosanitary status of the exporting country.

In terms of sustainability of the production systems, it is important to mention that all the disadvantages discussed in this research work, together with the scientific evidence that has proven the negative environmental impact loads generated by the existing production systems in places where large concentrations of greenhouses are established, have led to a significant reduction in the environmental impact of the production systems [50,61]. They provide a great opportunity for the search for sustainable strategies in the local context to be implemented in Colombian protected agriculture. It is possible to propose circularity strategies, strategies for the microclimatic optimization of the greenhouses analyzed. This is aimed at improving the yields of production systems, increasing the efficiency of irrigation and fertilization resources and promoting a lower use of chemical products for pest and disease control due to lower incidence and damage events [62]. 

Regarding the strategies for air conditioning, it should be mentioned that these should be based on the architectural redesign of the structures, increasing the ventilation surfaces above 40% of the covered area, seeking to maximize the renovation rates to values suitable for humid tropical conditions, guaranteeing renovation rates above 60 vol h^−1^ [63]. Strategies based on the use of photovoltaic agriculture or other renewable energy sources to be implemented for heating, cooling and dehumidification are another possibility widely implemented in recent years [64]. At the local level, there is also a lack of implementation and experimentation with passive technologies, such as other types of plastic coverings with special additives, such as anti-drip films to limit the effect of condensation, the use of shading nets to manage temperature and internal radiation at times of high solar radiation. For nighttime hours, possible solutions include the use of energy-saving thermal screens to limit thermal inversion phenomena, as well as the use of double roofs and green or plastic padding. Finally, feasibility studies and experimentation with passive thermal storage systems for heating purposes are also necessary. 

## 4. Conclusions

In this work, a microclimatic characterization of the five main types of greenhouses used in Colombia was carried out. As relevant results, it can be mentioned that in each of the structures evaluated, it was found that there is a direct relationship between the conditions of temperature, relative humidity and DPV generated inside the greenhouse and the level of external solar radiation.

It was found that by using 40 temperature and relative humidity sensors uniformly distributed in each of the structures and installed at the same level above ground level, it is possible to predict the spatial variability of temperature, relative humidity and VPD in each of the 5 greenhouses evaluated by using geostatistical techniques at unsampled points. Likewise, this spatial variability analysis allowed the heterogeneous microclimate conditions generated in this type of passive greenhouse to be identified; the mean nighttime temperature conditions varied for the same greenhouse interior environment from a minimum of 0.25 °C in HGT1 to a maximum of 4.1 °C in MG, while for daytime conditions, this value varied between a minimum of 2.7 °C in HGT1 and a maximum of 7.2 °C in SG.

In terms of relative humidity and VPD, it was found that there is also heterogeneity in the spatial behavior inside each greenhouse evaluated. In addition, it was identified that the dominant average conditions inside the greenhouses are high humidity and inadequate DPV values; this behavior can be a productive limiting factor of the crops and the main cause of loss of quality of the harvested products due to fungal diseases and attacks.

Due to these results, it is important to evaluate climate optimization strategies at the local level to improve the microclimatic behavior of the greenhouse structures used in Colombia; this will undoubtedly contribute to increasing the sustainability of the most relevant production systems in the country. Likewise, with the rise of agriculture 4.0 and the internet of things, climate monitoring practices and the use of microclimatic information collected for decision-making in agronomic crop management practices should be promoted among Colombian producers.

Finally, it is important to mention that methodologically, this work did not include cultivation variables and measurements related to plant growth, neither physical nor physiological variables. This was due to time constraints for the development of the microclimatic characterization of each of the greenhouses. However, in the future, complementary experimental studies can be proposed to monitor the growth and productivity of the main crops of ornamental or horticultural interest; under these types of fluctuating and heterogeneous protected environments. Likewise, these future studies will allow us to know the responses that plants can offer in the short and long term to environmental conditions since crops have acclimatization mechanisms that can influence physiological or agronomic responses. The above does not detract from the scientific importance of the research work carried out since the microclimate management of greenhouse crops requires the greatest possible knowledge and understanding on the part of decision-makers and producers themselves; therefore, identifying the relationships that exist between the microclimate generated in a type of structure and the local climatic conditions is key to the management of this type of greenhouse.

## Figures and Tables

**Figure 1 sensors-22-03925-f001:**
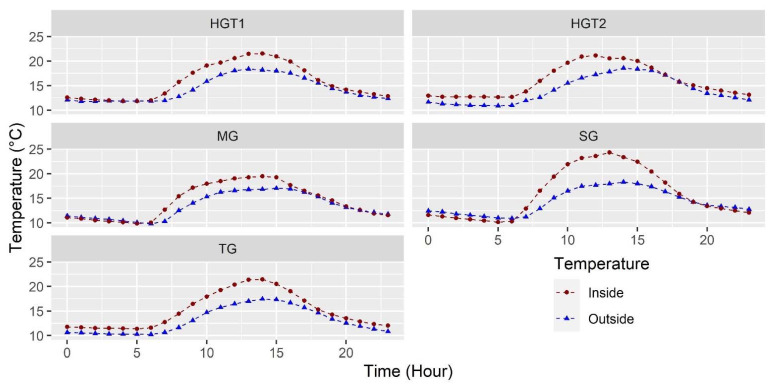
Mean temperature behavior for the 40 sensors in the indoor environment and mean outdoor temperature for each of the greenhouses evaluated.

**Figure 2 sensors-22-03925-f002:**
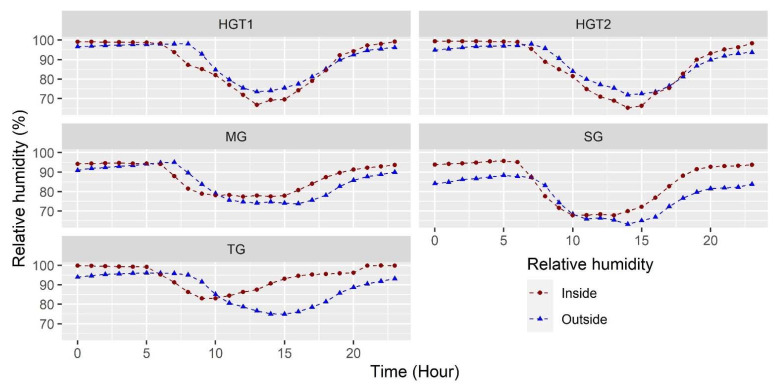
Mean relative humidity behavior for the 40 sensors in the indoor environment and mean outdoor relative humidity for each of the greenhouses evaluated.

**Figure 3 sensors-22-03925-f003:**
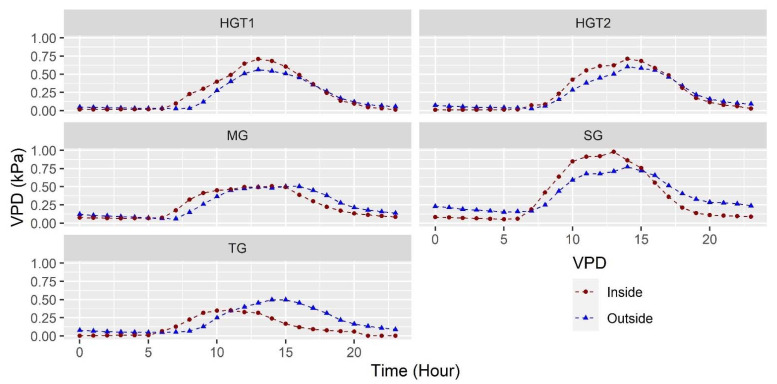
Mean VPD behavior for the 40 sensors in the indoor environment and mean outdoor VPD for each of the greenhouses evaluated.

**Figure 4 sensors-22-03925-f004:**
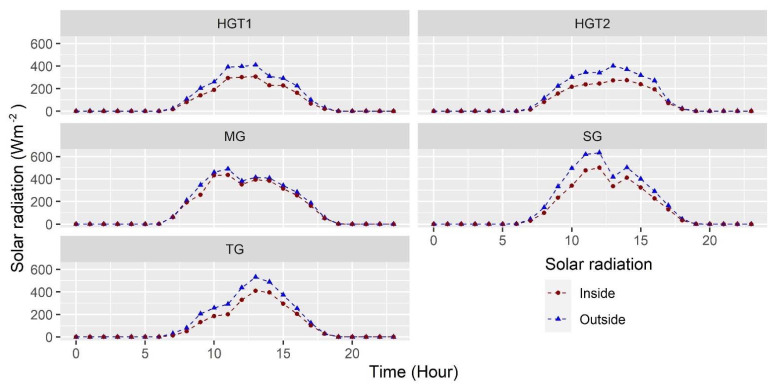
Mean radiation behavior for the sensor in the indoor environment and mean outdoor radiation solar for each of the greenhouses evaluated.

**Figure 5 sensors-22-03925-f005:**
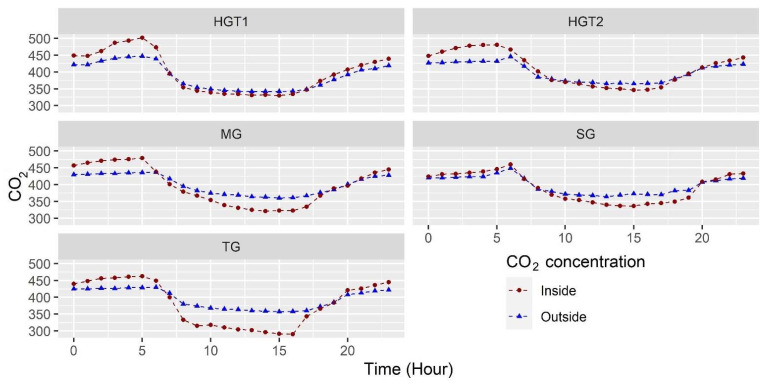
Mean CO_2_ concentration behavior for the sensor in the indoor environment and mean outdoor CO_2_ concentration for each of the greenhouses evaluated.

**Figure 6 sensors-22-03925-f006:**
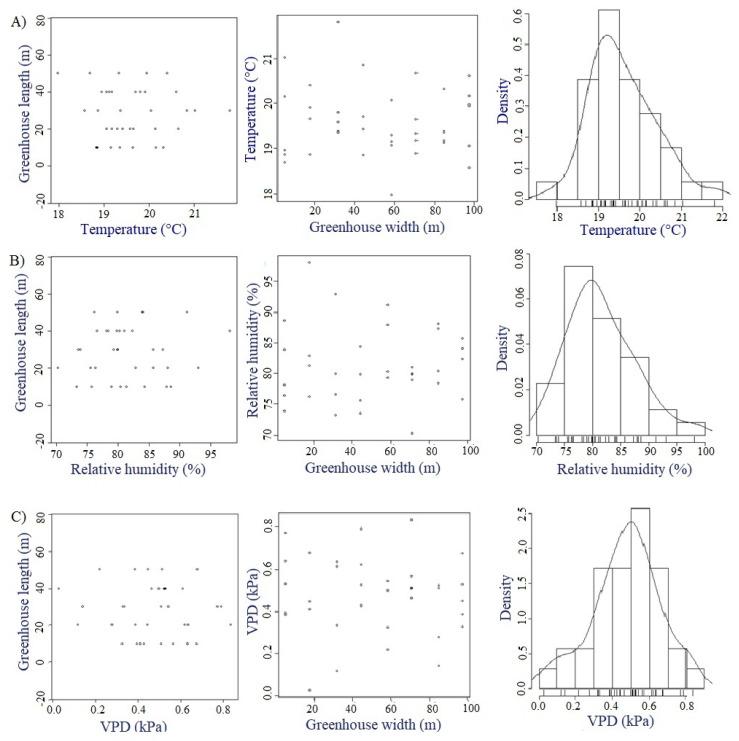
Results of exploratory spatial analysis, (**A**) Temperature in MG and ESRL 5, (**B**) Relative humidity in MG and ESRL 3 and (**C**) VPD in MG and ESRL 4.

**Figure 7 sensors-22-03925-f007:**

Spatial distribution predicted by kriging, for temperature, relative humidity and VPD in MG and for ESRL 1.

**Figure 8 sensors-22-03925-f008:**
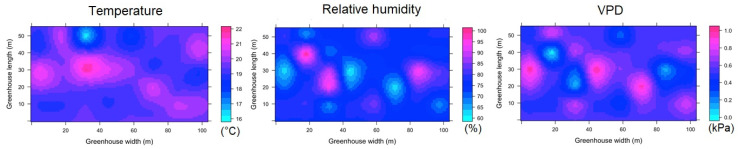
Spatial distribution predicted by kriging, for temperature, relative humidity and VPD in MG and for ESRL 5.

**Figure 9 sensors-22-03925-f009:**
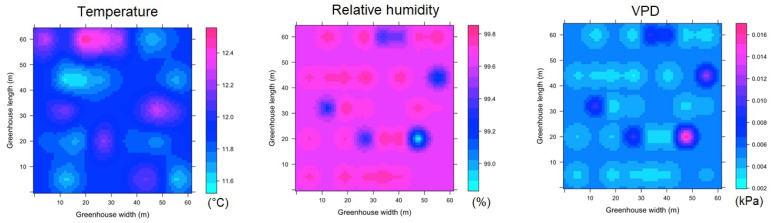
Spatial distribution predicted by kriging, for temperature, relative humidity and VPD in TG and for ESRL 1.

**Figure 10 sensors-22-03925-f010:**
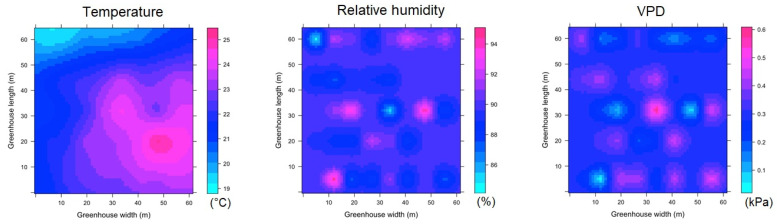
Spatial distribution predicted by kriging, for temperature, relative humidity and VPD in TG and for ESRL 5.

**Figure 11 sensors-22-03925-f011:**
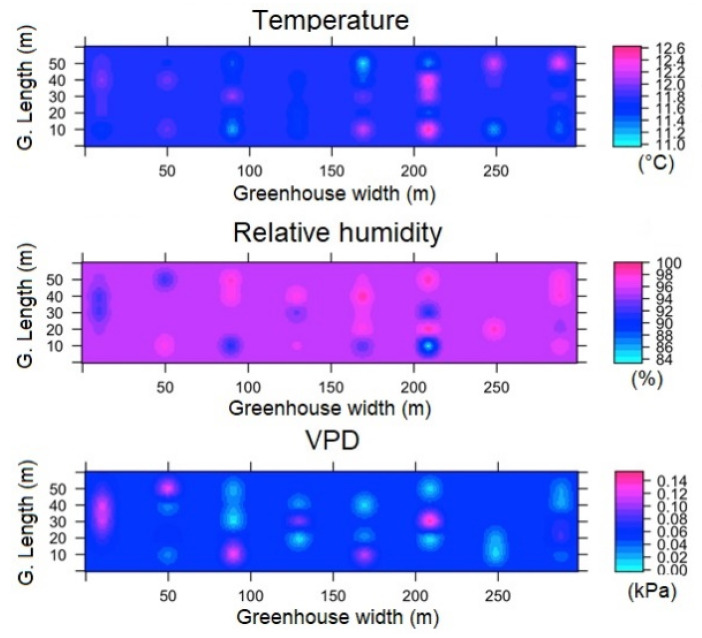
Spatial distribution predicted by kriging, for temperature, relative humidity and VPD in SG and for ESRL 1.

**Figure 12 sensors-22-03925-f012:**
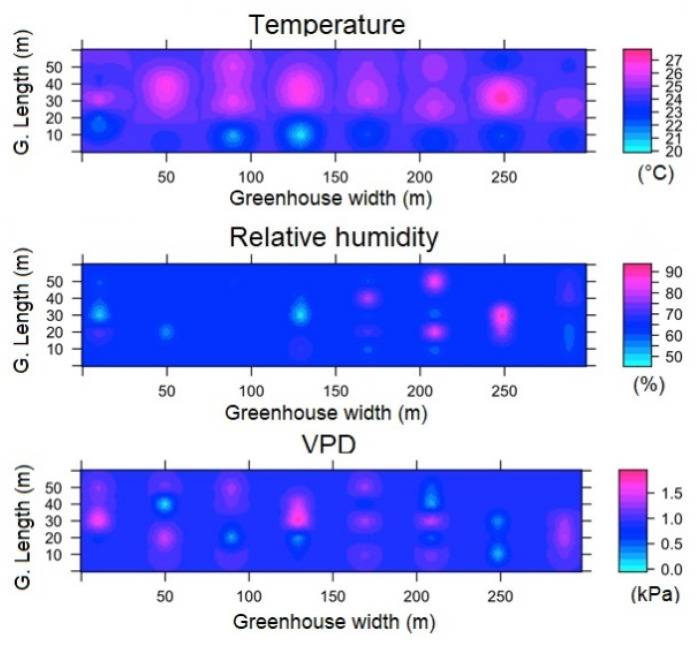
Spatial distribution predicted by kriging, for temperature, relative humidity and VPD in SG and for ESRL 5.

**Figure 13 sensors-22-03925-f013:**
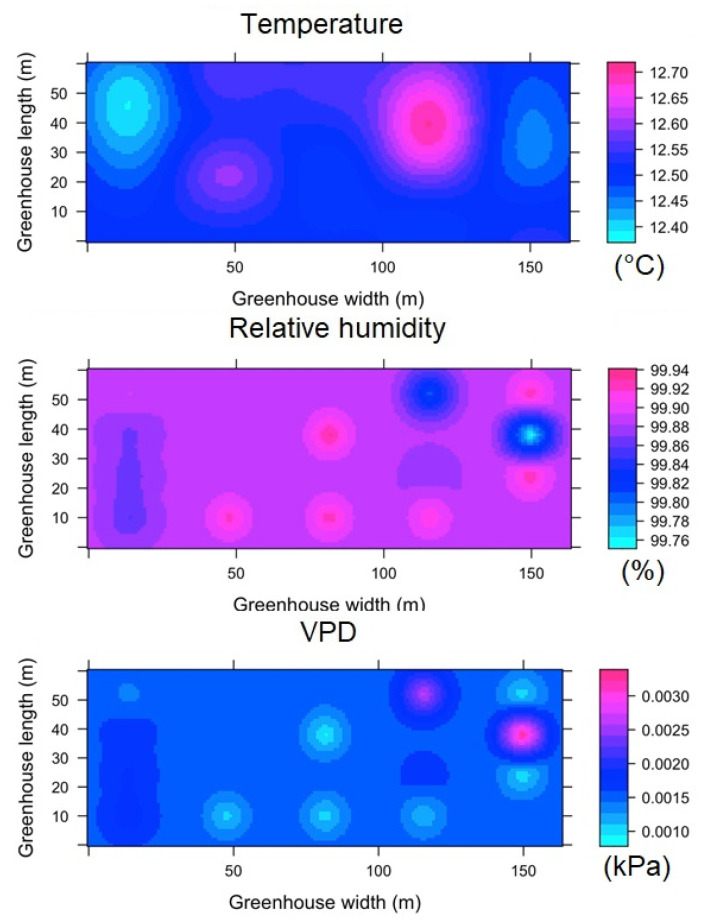
Spatial distribution predicted by kriging, for temperature, relative humidity and VPD in HGT1 and for ESRL 1.

**Figure 14 sensors-22-03925-f014:**
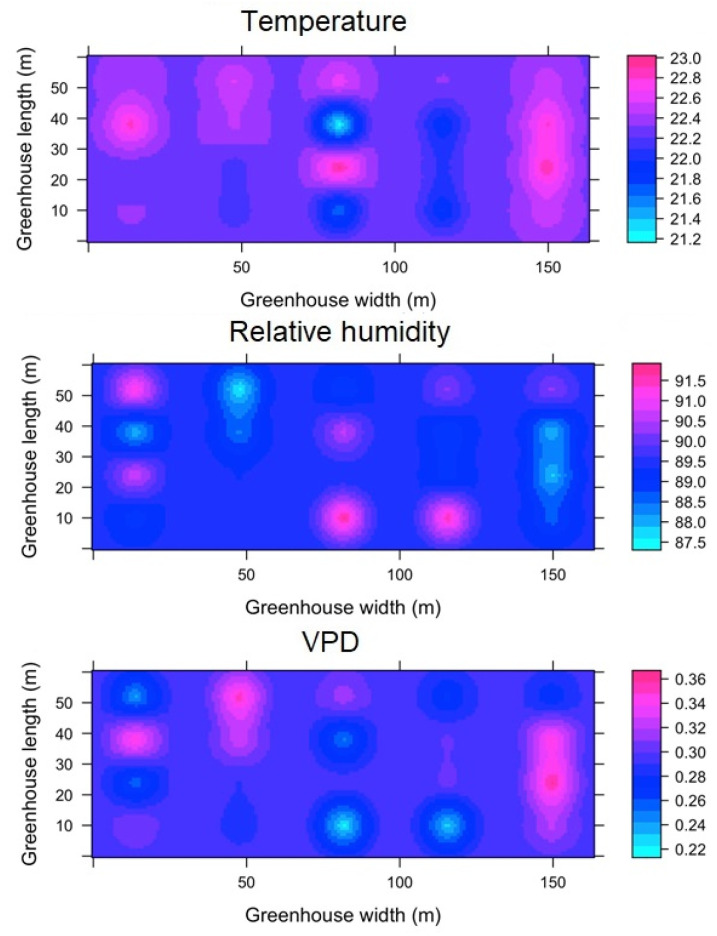
Spatial distribution predicted by kriging, for temperature, relative humidity and VPD in HGT1 and for ESRL 5.

**Figure 15 sensors-22-03925-f015:**
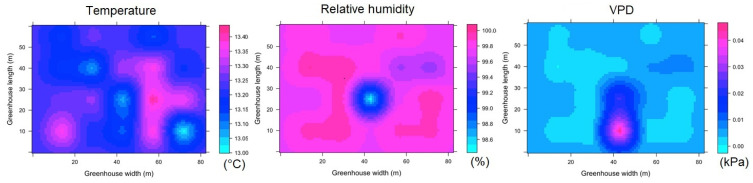
Spatial distribution predicted by kriging, for temperature, relative humidity and VPD in HGT2 and for ESRL 1.

**Figure 16 sensors-22-03925-f016:**
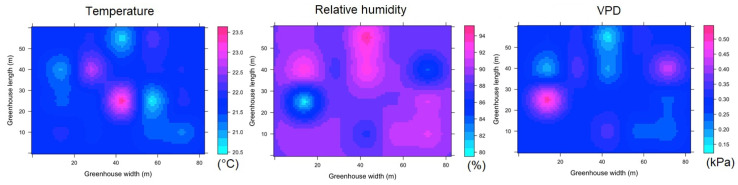
Spatial distribution predicted by kriging, for temperature, relative humidity and VPD in HGT2 and for ESRL 5.

**Table 1 sensors-22-03925-t001:** General characteristics of the TG greenhouse.

Greenhouse Schematic	Description
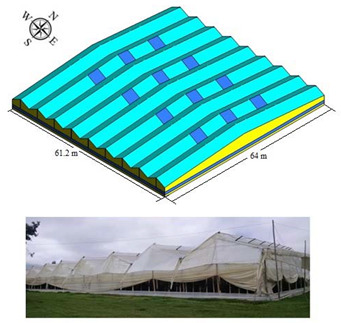	Model TG Number of spans (n): 9 Width of span (m): 6.8 Total width (m): 61.2 Greenhouse length (m): 64 Surface area covered (Sac, m^2^): 3916.8 Rooftop ventilation area (m^2^): 134.4 Side ventilation area (m^2^): 128 Front ventilation area (m^2^):122.4 Total ventilation surface area (Tvsa, m^2^): 3848 Ventilation ratio (Tvsa/Sac, %): 10.17 Minimum height over gutter (m): 3.3 Minimum height above roof (m): 5.2 Maximum height over gutter (m): 6.1 Maximum height above roof (m): 8.0

**Table 2 sensors-22-03925-t002:** General characteristics of the MG greenhouse.

Greenhouse Schematic	Description
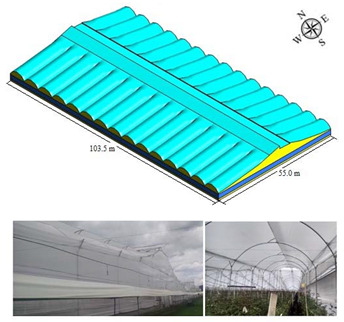	Model MG Number of spans (n): 15 Width of span (m): 6.9 Total width (m): 103.5 Greenhouse length (m): 55 Surface area covered (Sac, m^2^): 5692.5 Rooftop ventilation area (m^2^): 313.5 Side ventilation area (m^2^): 220 Front ventilation area (m^2^): 414 Total ventilation surface area (sv, m^2^): 9475 Ventilation ratio (Tvsa/Sac, %): 16.64 Minimum height over gutter (m): 3.1 Minimum height above roof (m): 5.5 Maximum height over gutter (m): 5.4 Maximum height above roof (m): 8.2

**Table 3 sensors-22-03925-t003:** General characteristics of the HGT1 greenhouse.

Greenhouse Schematic	Description
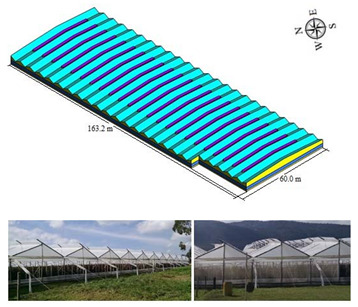	Model HGT1 Number of spans (n): 24 Width of span (m): 6.8 Total width (m): 163.2 Greenhouse length (m): 60 Surface area covered (Sac, m^2^): 9656 Rooftop ventilation area (m^2^): 1507.2 Side ventilation area (m^2^): 228 Front ventilation area (m^2^): 620.16 Total ventilation surface area (Tvsa, m^2^): 2355.36 Ventilation ratio (Tvsa/Sac, %): 24.39 Minimum height over gutter (m): 3.0 Minimum height above roof (m): 5.2 Maximum height over gutter (m): 5.2 Maximum height above roof (m): 7.4

**Table 4 sensors-22-03925-t004:** General characteristics of the HGT2 greenhouse.

Greenhouse Schematic	Description
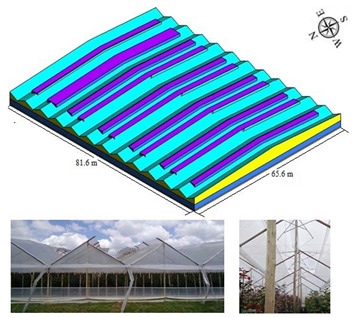	Model HGT2 Number of spans (n): 12 Width of span (m): 6.8 Total width (m): 81.6 Greenhouse length (m): 65.6 Surface area covered (Sac, m^2^): 5352.9 Rooftop ventilation area (m^2^): 942.72 Side ventilation area (m^2^): 236.16 Front ventilation area (m^2^): 293.76 Total ventilation surface area (Tvsa, m^2^): 1472.64 Ventilation ratio (Tvsa/Sac, %): 27.51 Minimum height over gutter (m): 3.0 Minimum height above roof (m): 5.4 Maximum height over gutter (m): 5.4 Maximum height above roof (m): 7.8

**Table 5 sensors-22-03925-t005:** General characteristics of the SG greenhouse.

Greenhouse Schematic	Description
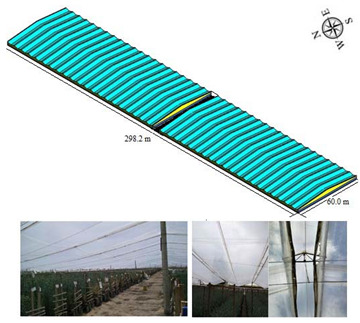	Model SG Number of spans (n): 42 Width of span (m): 7.1 Total width (m): 298.2 Greenhouse length (m): 60 Surface area covered (Sac, m^2^): 17.892 Rooftop ventilation area (m^2^): 615 Side ventilation area (m^2^): 264 Front ventilation area (m^2^): 1312.1 Total ventilation surface area (Tvsa, m^2^): 2191.1 Ventilation ratio (Tvsa/Sac, %): 12.24 Minimum height over gutter (m): 3.5 Minimum height above roof (m): 5.5 Maximum height over gutter (m): 5.5 Maximum height above roof (m): 7.5

**Table 6 sensors-22-03925-t006:** Climate monitoring sensors used in the microclimatic characterization of the 5 greenhouses.

Sensors Used Outside	Sensors Used Inside Greenhouses
1 automatic weather station i-Metos Compact station (Pessl Instruments GmbH, Weiz, Austria). Data recording of temperature (°C), air humidity (%), wind speed (ms^−1^), direction (°), solar radiation (Wm^−2^) and precipitation (mm). 1 Autopilot Apcem 2 air CO_2_ concentration sensor (ppm) (Hydrofarm, China).	40 type-T copper thermocouples (Copper/Constantan) used to measure temperature and relative humidity. 40 data recorder (Cox-Tracer Junior, Escort DLS, Edison, NJ, EE. UU.). 1 Autopilot Apcem 2 air CO_2_ concentration sensor (ppm) (Hydrofarm, China). 1 pyranometer Li-Cor LI-200SZ (LI-COR Inc, EE. UU.)

**Table 7 sensors-22-03925-t007:** Periods of experimental evaluation in each of the greenhouses.

Greenhouse	Measurement Period
HGT1	1 February to 10 March.
HGT2	12 March to 22 April.
TG	24 April to 30 May.
MG	1 June to 9 July.
SG	12 July to 22 August.

**Table 8 sensors-22-03925-t008:** Radiation levels in which the data collected in the 5 greenhouses were grouped.

ESRL	ESRL 1	ESRL 2	ESRL 3	ESRL 4	ESRL 5
Solar radiation level (Wm^−2^).	[0, 0]	[0.1–91.8]	[91.9–274.1]	[274.2–486.3]	[486.4–968.8]

**Table 9 sensors-22-03925-t009:** Model parameters of the selected theoretical semivariograms and their quality of fit measures (Akaike Information Criteria; AIC, and Bayesian; BIC) for the temperature, relative humidity and VPD datasets of the five greenhouses evaluated.

	Temperatura (°C)								Humedad Relativa (%)							VPD (kPa)						
GT	ESRL	Modelo	*c* _0_	*c*_1_ (°C)	(*c*_0_*/c*_1_) (%)	*a* (m)	AIC	BIC	Modelo	*c* _0_	*c*_1_ (%)	(*c*_0_*/c*_1_) (%)	*a* (m)	AIC	BIC	Modelo	*c_0_*	*c*_1_ (kPa)	(*c*_0_*/c*_1_) (%)	*a* (m)	AIC	BIC
TG	1	Circular	0.08	0.49	16.3	12.08	46.6	52.8	Circular	0.01	0.121	8.2	6.67	57.9	63.7	Circular	0.00	0.0005	0.0	6.6	−213.4	−207.6
TG	2	Circular	0.00	0.07	0.0	13.12	35.4	41.6	Circular	0.00	0.48	0.0	3.75	116.7	122.4	Circular	0.00	0.0002	0.0	5.0	−140.0	−134.3
TG	3	Esférico	0.07	0.38	18.4	69.54	44.8	51.1	Circular	0.00	15.12	0.0	7.21	196.8	202.9	Circular	0.00	0.007	0.0	7.2	−66.2	−60.1
TG	4	Esférico	0.00	1.13	0.0	19.58	108.5	114.9	Circular	0.00	14.24	0.0	7.26	194.7	200.9	Circular	0.00	0.0095	0.0	8.0	−53.8	−47.7
TG	5	Circular	0.37	2.90	12.7	64.18	122.1	128.2	Circular	6.59	29.69	22.1	6.62	199.3	205.4	Circular	0.00	0.0165	0.0	7.9	−35.2	−29.1
MG	1	Circular	0.02	0.13	15.3	9.97	125.1	131.6	Circular	0.0	7.7	0.0	12.9	168.9	174.9	Circular	0.0	0.001	0.0	12.9	−115.5	−109.5
MG	2	Circular	0.00	0.18	0.00	13.42	48.4	54.6	Circular	0.0	10.5	0.0	13.6	178.8	184.8	Circular	0.0	0.003	0.0	13.7	−96.3	−90.3
MG	3	Circular	0.06	0.34	17.6	9.45	94.7	101.7	Circular	1.7	20.5	8.3	9.56	232.0	238.2	Circular	0.001	0.028	3.5	9.7	−41.3	−35.1
MG	4	Esférico	0.11	0.59	18.6	9.88	99.4	105.9	Circular	0.0	65.7	0.0	12.6	253.5	259.7	Circular	0.0	0.033	0.0	12.5	−12.2	−5.9
MG	5	Circular	0.00	0.92	0.00	15.39	108.4	114.9	Circular	0.0	84.3	0.0	12.9	262.2	268.4	Circular	0.0	0.052	0.0	12.9	3.2	9.4
SG	1	Circular	0.02	0.16	15.3	9.83	125.1	54.9	Circular	2.8	14.1	20.1	9.97	166.7	172.0	Circular	0.00	0.002	0.0	11.5	−101.8	−95.8
SG	2	Circular	0.10	0.90	11.1	19.48	48.4	52.2	Circular	2.5	22.5	11.4	9.99	163.4	168.7	Circular	0.00	0.005	0.0	9.94	−70.7	−64.8
SG	3	Circular	0.00	0.52	0.0	20.40	94.7	88.3	Circular	6.9	38.7	18.0	9.82	194.3	199.8	Circular	0.00	0.034	0.0	9.85	−5.8	0.3
SG	4	Circular	0.00	1.70	0.0	20.76	99.4	135.0	Circular	5.4	58.7	9.2	9.99	210.7	216.2	Circular	0.005	0.11	4.5	10.9	27.0	33.2
SG	5	Circular	0.00	3.10	0.0	20.65	147.7	154.1	Circular	0.0	114.1	0.0	10.8	227.6	233.1	Circular	0.0	0.22	0.0	12.9	53.1	59.4
HGT1	1	Gaussiano	0.02	0.12	16.6	105.9	22.6	26.6	Gaussiano	0.00	0.00	0.0	3.54	−20.47	−17.6	Circular	0.0	0.0	0.0	35.94	−137.4	−134.4
HGT1	2	Gaussiano	0.02	0.12	16.6	26.16	17.8	21.8	Gaussiano	0.05	0.23	21.7	22.6	12.66	15.7	Gaussiano	0.0	0.0	0.0	23.06	−118.4	−115.3
HGT1	3	Circular	0.00	0.20	0.0	28.91	30.1	34.1	Gaussiano	0.07	0.41	17.1	3.44	39.31	42.4	Circular	0.0	0.0	0.0	13.44	−81.3	−78.0
HGT1	4	Circular	0.08	0.53	15.1	26.40	53.3	57.2	Gaussiano	0.06	0.28	21.4	16.7	27.37	28.5	Circular	0.0	0.002	0.0	17.40	−54.7	−51.3
HGT1	5	Circular	0.07	0.90	7.1	18.85	64.0	68.0	Circular	0.01	4.12	0.24	17.0	80.31	83.6	Circular	0.0001	0.003	3.3	17.85	−41.1	−37.8
HGT2	1	Gaussiano	0.04	0.25	16.0	19.91	2.4	6.6	Gaussiano	0.0	0.001	0.0	8.34	−50.8	−47.2	Circular	0.0	0.0	0.0	14.09	−203.2	−199.6
HGT2	2	Circular	0.0	0.053	0.0	14.09	5.9	9.7	Circular	0.0	0.046	0.0	14.09	3.5	7.1	Circular	1.88	0.0	0.0	14.09	−143.2	−139.5
HGT2	3	Circular	0.0	0.115	0.0	14.09	21.4	25.4	Circular	0.0	0.645	0.0	14.09	51.1	54.7	Circular	0.0	0.0	0.0	14.09	−85.5	−81.95
HGT2	4	Circular	0.0	0.344	0.0	14.09	43.4	47.4	Circular	0.0	2.268	0.0	14.09	77.4	81.2	Circular	0.0	0.002	0.0	14.09	−61.0	−57.3
HGT2	5	Circular	0.0	0.67	0.0	14.09	56.7	60.7	Circular	0.0	5.048	0.0	14.09	92.6	96.4	Circular	0.0	0.005	0.0	14.09	−40.2	−36.7

*c*_0_; nugget effect, *c*_1_: sill parameter, *a*; range, AIC; Akaike Information Criteria, BIC; Bayesian Information Criteria.

## Data Availability

Not applicable.

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
