# Peer review of "Microclimatic Evaluation of Five Types of Colombian Greenhouses Using Geostatistical Techniques"

_sensors, 2022, doi:10.3390/s22103925_

Round 1
Reviewer 1 Report
The manuscript "Microclimatic Evaluation of Five Types of Colombian Greenhouses Using Geostatistical Techniques" shows an analysis of the climatic conditions in 5 types of greenhouses in Colombia through a geostatistical analysis. The proposed study is not innovative. Moreover, it is well known that there is heterogeneity of climatic conditions in greenhouses. The authors proposed neither microclimatic optimization strategies nor overlap between climatic data and yields. At last, the discussion of the results and the comparison with other similar experiences is completely missing. For this reason the work cannot be accepted for publication.
Other comments/suggestions follow.
Throughout the paper, put one decimal place for temperatures, zero decimal places for RH and solar radiation, two decimal places for VPD.
Line 53: Please, modify “[10]” with “[10].”
Line 137: Please, modify “hanging” with “Hanging”
Line 184: Please, enter the equation for VPD
Lines 195-197: Please, explain how the radiation level classes were identified
Line 198: Please, modify “DPV” with “VPD”
Line 228: There is not Mph in equation 1
Line 231-232: Gaussian, was repeated 2 times. The fifth theoretical model is missing
Line 266 : Are these average temperatures for the period under consideration? If so, it must be said
Lines 269-270: Not quite so in the picture. In fact, the figures show the phenomenon of thermal hysteresis
Figures 1-5: all are assumed to be averaged data over the 40-day period. If so, it must be said
Author Response
Dear reviewer, we thank you for your constructive comments, we have included most of them in the new version of the document. We have also attached a response file where you can find all the modified details.

Reviewer 2 Report
The Authors describe different types of greenhouses, arguing that each type has its own specific characteristics and properties that directly affect the phenomenon of natural ventilation. The Authors diagnose a negative aspect among Colombian producers, which is the lack of a culture of using appropriate tools to correlate microclimatic data with production data. The authors indicate methods of production optimization through the use of computer simulation tools and multidimensional information analysis, taking into account a large number of measurement variables for 5 types of greenhouses. Interesting, well-described and visualized research.
Comments - too many self-citations - should be reduced. In the abstract, the Authors refer to the term sustainability, which appears only once in the main text - the Authors do not explain how the proposed solutions will affect sustainability. This thread should be expanded.
Best regards
Reviewer
Author Response

(The authors gave the same response as above.)

Reviewer 3 Report
I advise the authors to consider addressing the following comments:
- It is reasonable that a modern greenhouse will have better cultivation results. A substantial cost, however, is required to achieve this. The question that emerges is whether Colombian farmers can cover this cost. The authors do not provide any relevant information. Instead, it is stated that it is urgent to propose microclimatic optimization strategies to help ensure the sustainability of the most important production systems in the country.
- The section of Methodology must be supported with the appropriate references.
- It should also be explained why the specific variables were chosen to be examined.
- In addition, the authors must clarify whether the examined variables affect in the same way the development of cultivated plants in the plant nurseries of Colombia.
- The authors should also explain whether the cost required for greenhouse upgrading can be covered by the expected yield increase.
- Sentences in lines 614-615 convey important information which should be supported and referenced with the appropriate references.
- The paper is overall well-written, however, a minor check for typos could be beneficial.
Author Response
Dear reviewer, thank you for your constructive comments, we have included most of them in the new version of the document. We have also attached a response file where you can find all the modified details.

Round 2
Reviewer 1 Report
The authors accepted the reviewers' suggestions, improving the quality of the manuscript. However, it would have been more complete and interesting to include parameters such as yield, etc., as suggested during the first review. At this point, I invite the authors to investigate these aspects, because this would really give practical and quantifiable indications to end users.
At last, I suggest moving lines 283-297, in the "Conclusions" section, adapting the sentences for this purpose.
Author Response
Bogota, May 17, 2022.
Dear Reviewer.
Thank you very much for your response and advice. We agree with you that we must deepen the research related to crop variables, we are waiting to start a specific project on rose and alstroemeria where we will follow up these crops and perform physical and physiological measurements.
Finally, we have moved the text suggested by you to the conclusions section.
Regards and our best wishes.
The authors.